# OpenReview forum: "Decoupled Alignment for Robust Plug-and-Play Adaptation"
_ICLR.cc/2025/Conference — ICLR 2025 Conference Withdrawn Submission_

### Official Review · Reviewer_Z3rx · 2024-10-29

**Soundness:** 2
**Presentation:** 3
**Contribution:** 2
**Rating:** 3
**Confidence:** 4

**Summary:**

This paper proposes DAPA, an LLM safety enhancement method without SFT/RLHF. This is performed by first identifying the "alignment knowledge" stored in the MLP module of an aligned model, and transfer the knowledge to unaligned models. Empirically, the defense success rate was improved, and the utility was not sacrificed.

**Strengths:**

The tackled problem, LLM safety and jailbreak, is timely. The proposed delta debugging algorithm for finding alignment knowledge within the MLP module is novel, and the ablation study as well as utility test is comprehensive.

**Weaknesses:**

The main weaknesses I consider are as follows.
- DAPA requires an aligned LLM. Although claimed as "a low-resource safety enhancement method for aligning LLMs", the preliminary of DAPA algorithm for aligning an unaligned LLM is **the very existence** of an aligned LLM. This makes the stated contribution vague since it cannot serve as a substitution to SFT/RLHF, and thus cannot reduce the resource used for alignment or be an "economic solution to LLM safety enhancement".
- The safety test is weak. Throughout the paper, the defense success rate is measured using AdvBench **directly**. However, given the rapid development of LLM safety (or specifically, jailbreak), there are a suite of jailbreak adversaries proposed and been collected by some benchmark projects (e.g., [1] [2]). Thus, it is insufficient to claim safety improvement without testing the model performance against **any** of these representative adversaries.
- The claim of "the alignment knowledge is stored in the Feed-Forward Network (FFN) layers" lacks supportive evidence. This claim is deduced in the paper via Figure 3, where only **a single harmful prompt** is utilized to compare the influence of the MLP module and the attention module across layers. More quantitative results are needed to support this result.

[1] HarmBench: A Standardized Evaluation Framework for Automated Red Teaming and Robust Refusal
[2] JailbreakBench: An Open Robustness Benchmark for Jailbreaking Large Language Models

**Questions:**

My questions to authors are as follows.
- Among the unaligned models, there are several base candidates that have not been tuned to follow instructions. Have you launched additional instruction following on non-harmful datasets in addition to the relative memory substitution described in section 4.1? If not, the result with base models will be hard to compare against instruction-tuned ones.
- When extending Figure 3 to incorporate multiple harmful prompts, is it still true that the MLP's influence is the strongest?
- Can DAPA maintain its benefit when encountering more advanced jailbreak adversaries?

---

> ### Author Response · Authors · 2024-11-20
>
> >**Reviewer's Comment:** DAPA requires an aligned LLM. Although claimed as "a low-resource safety enhancement method for aligning LLMs", the preliminary of DAPA algorithm for aligning an unaligned LLM is the very existence of an aligned LLM. This makes the stated contribution vague since it cannot serve as a substitution to SFT/RLHF, and thus cannot reduce the resource used for alignment or be an "economic solution to LLM safety enhancement".
>
> **Response:**
>
> Thank you for your thoughtful critique. We understand your concern about the dependency of DAPA on an aligned LLM and how this may seem to conflict with our claim of being a low-resource solution for safety enhancement.
>
> The key distinction we aim to highlight in our work is that while DAPA leverages an existing aligned model, it does so in a way that significantly reduces the need for costly alignment processes like supervised fine-tuning (SFT) or reinforcement learning from human feedback (RLHF). In conventional alignment strategies, each unaligned model would need to undergo extensive, resource-intensive training to achieve alignment. In contrast, DAPA allows the alignment knowledge to be transferred efficiently from one well-aligned model to multiple unaligned models with minimal parameter updates (on average only 6.26% of parameters), effectively reducing computational costs and resources.
>
> It is important to note that DAPA is not intended to replace the initial alignment of the teacher model itself. Instead, it serves as a scalable and efficient method to propagate alignment knowledge once a reliable teacher model is available. Thus, while it relies on the existence of an initially aligned LLM, DAPA provides a practical solution to align a broader range of models without the need for repeated, resource-intensive fine-tuning or RLHF processes for each model.
>
> It is also worth noting that while SFT-aligned models incur significant computational costs, they often fail to demonstrate robust defense against jailbreak attacks, as highlighted in [1][2]. In contrast, DAPA achieves meaningful improvements in Defense Success Rate (DSR) with minimal computational overhead by focusing on targeted memory editing rather than full model retraining. This positions DAPA as a complementary approach that enhances the utility of existing aligned models while addressing the robustness and efficiency trade-offs inherent in traditional methods.
>
> In summary, our contribution is in providing a method to economically extend alignment from one model to many, rather than claiming to eliminate the need for the initial alignment of the teacher model. We clarify this point in the revised version to ensure that our contribution is accurately conveyed. Thank you for bringing this to our attention.
>
> 【1】Jingwei Yi, Rui Ye, Qisi Chen, Bin Zhu, Siheng Chen, Defu Lian, Guangzhong Sun, Xing Xie, and Fangzhao Wu. 2024. [On the Vulnerability of Safety Alignment in Open-Access LLMs](https://aclanthology.org/2024.findings-acl.549). In Findings of the Association for Computational Linguistics: ACL 2024, pages 9236–9260, Bangkok, Thailand. Association for Computational Linguistics.
>
> 【2】 Qi, Xiangyu, et al. "Fine-tuning aligned language models compromises safety, even when users do not intend to!." arXiv preprint arXiv:2310.03693 (2023).
>
> >**Reviewer's Comment:** The safety test is weak. Throughout the paper, the defense success rate is measured using AdvBench directly. However, given the rapid development of LLM safety (or specifically, jailbreak), there are a suite of jailbreak adversaris proposed and been collected by some benchmark projects (e.g., [1] [2]). Thus, it is insufficient to eclaim safety improvement without testing the model performance against any of these representative adversaries.
>
> **Response:**
>
> We appreciate the reviewer’s insightful feedback regarding the safety evaluation in our paper. We acknowledge this concern and have expanded our evaluation in the revised paper. In addition to AdvBench, we also tested DAPA on two additional datasets: HarmfulQA (Appendix G.1) and JailbreakBench (Appendix G.7).
>
> -   HarmfulQA: DAPA achieved an average DSR improvement of 8.02%, with improvements reaching up to 15% in some cases across 17 LLM models.
>
> -   JailbreakBench: Incorporating this dataset, as suggested, we observed a 3.1% improvement in DSR, validating DAPA's effectiveness against diverse jailbreak-style adversarial prompts.
>
>
> These results provide a more comprehensive assessment of LLM safety, covering a wider variety of adversarial scenarios. We appreciate your valuable suggestion and have included these updates in the revised manuscript.

---

> ### Author Response · Authors · 2024-11-20
>
> >**Reviewer's Comment:** The claim of "the alignment knowledge is stored in the Feed-Forward Network (FFN) layers" lacks supportive evidence. This claim is deduced in the paper via Figure 3, where only a single harmful prompt is utilized to compare the influence of the MLP module and the attention module across layers. More quantitative results are needed to support this result. When extending Figure 3 to incorporate multiple harmful prompts, is it still true that the MLP's influence is the strongest?
>
> **Response:**
>
> Thank you for your question about the evidence supporting our claim regarding alignment knowledge storage in FFN/MLP layers. We acknowledge your concern about our initial single-prompt experiment's limitations and have addressed this by conducting additional experiments using multiple harmful prompts. These experiments include additional visualizations of the memory space, as shown in Appendix G.12.
>
> Our results indicate that the hidden states in the middle layers of the model have the most significant impact on the model’s output. Moreover, the heatmap shows that MLP layers have a higher indirect effect than the Attention and ALL modules. Notably, we also observe that the ALL module closely aligns with the pattern observed in the MLP layers further supporting MLP's dominance in storing alignment knowledge.
>
> This observed pattern remains consistent across different prompt types, suggesting that it reflects an architectural feature rather than a phenomenon tied to specific prompts. These findings provide robust support for our original claim regarding the role of MLP layers in alignment knowledge storage.
>
> >**Reviewer's Comment:** Among the unaligned models, there are several base candidates that have not been tuned to follow instructions. Have you launched additional instruction following on non-harmful datasets in addition to the relative memory substitution described in section 4.1? If not, the result with base models will be hard to compare against instruction-tuned ones.
>
> **Response:**
>
> Thank you for your insightful question. In our experiments, we did not perform additional instruction tuning on the base models beyond the relative memory substitution described in Section 4.1. The focus of our study was to evaluate DAPA's effectiveness in transferring alignment knowledge from teacher models to unaligned models without additional fine-tuning.
>
> We acknowledge that comparing base models directly against instruction-tuned ones may introduce discrepancies due to differences in their initial capabilities. To address this, we conducted additional experiments to evaluate the robustness of DAPA on instruction-tuned foundation models. Specifically, we utilized the ShareGPT unfiltered dataset for supervised instruction-tuned fine-tuning. Using the QLORA method, we fine-tuned the Llama2-7B model with the Llama2-7B-chat template, conducting training on two NVIDIA A100 80G GPUs over 15,000 steps. After alignment with DAPA, the model was tested on AdvBench, where the Defense Success Rate (DSR) improved from 10.16% to 18.4%.
>
> Additionally, in our original experiments, the same chat template was used for both pre- and post-DAPA models, ensuring a fair comparison of performance and safety. This consistent setup ensures that the improvements observed after applying DAPA are not influenced by differences in model architecture or templates, but rather stem directly from our alignment method.
>
> These results validate DAPA’s effectiveness in improving both performance and safety under fair experimental conditions. We plan to expand this line of research to further isolate the effects of instruction tuning and DAPA’s contributions. Thank you for your suggestion, and we will incorporate these findings into the revised paper for a more comprehensive evaluation.
>
> >**Reviewer's Comment:** Can DAPA maintain its benefit when encountering more advanced jailbreak adversaries?
>
> **Response:**
>
> We appreciate your question regarding DAPA’s effectiveness against more advanced jailbreak adversaries. To evaluate this, we tested DAPA on AdvBench using **GPTFuzzer**, a state-of-the-art jailbreak tool, against Llama-2-7b models.
>
> As shown in Appendix G.8, the histogram illustrates the Defense Success Rate (DSR) of the **Llama-2-7b-hf** model. Before alignment, the DSR was **19%**, whereas after applying DAPA, the DSR significantly increased to **31%**. These results demonstrate DAPA's robustness and adaptability, maintaining its advantage even when encountering sophisticated jailbreak techniques. This highlights the method's potential for enhancing model safety in dynamic and challenging environments.

---

> > ### Author Response · Authors · 2024-11-22
> >
> > Thank you again for the review. Your careful reading and insightful comments indeed help us a lot in improving our work. Since the discussion phase is about to end, we are writing to kindly ask if you have any additional comments regarding our response. In addition, if our new experiments address your concern, we would like to kindly ask if you could consider raising the score.

---

> > ### Comment · Reviewer_Z3rx · 2024-11-25
> > **Official Comment by Reviewer Z3rx**
> >
> > I thank the authors for the detailed response and for updating the paper.
> >
> > - On requirement of an aligned LLM. I understand the purpose of DAPA is to reduce the alignment cost by **transferring knowledge** from a teacher model, however, in this case, it is improper to call the proposal an alignment method "without the need for SFT or RLHF", since the dependence is implicit. I would suggest the author to be more careful on the description of the contribution made with this proposal.
> >
> > - On safety evaluations and more advanced adversaries. I thank the authors for conducting additional experiments on new datasets, and on GPTFuzzer, a more advanced adversary. These results do support the effectiveness of DAPA, however, the limitation of lacking strong enough adversaries still exist. By adversary, I am referring to **methods** such as GPTFuzzer and the list of candidates considered in e.g., HarmBench [1]. The inclusion of HarmfulQA and (direct, non-artifact) JailbreakBench still falls into the direct prompting regime and thus does not provide additional information on capabilities against advanced adversaries.
> >
> > - On FFN knowledge storage. I thank the authors for the additional figures, however, the inclusion of **4** more prompts does not address the concern - for example, one of the provided prompt in Figure 10 suggests the last layer MLPs are the most important components. To make the claim faithful, a statistical evaluation with a reasonable number of prompts considered should be provided. I acknowledge that given the experimental results, it is reasonable to consider MLPs as the main objective of study, I am arguing there lacks support for this claim.
> >
> > -  On base model candidates. I thank the authors for providing additional fine-tuning results on Llama2-7B. However, the concern is around the **faithfulness of base model results** presented in the main text, since they are not trained to follow instructions. Thus, the defense success rate of these models will have a strong dependence on how the input format is designed to make sure the model output is reasonable. The robustness to the format is not presented in the paper, making the "improvement" statement hard to interpret.
> >
> > Given the above concerns, I decide to maintain my score.
> >
> > [1] HarmBench: A Standardized Evaluation Framework for Automated Red Teaming and Robust Refusal

---

> > > ### Author Response · Authors · 2024-11-26
> > >
> > > >**Reviewer's Comment:** On requirement of an aligned LLM. I understand the purpose of DAPA is to reduce the alignment cost by transferring knowledge from a teacher model, however, in this case, it is improper to call the proposal an alignment method "without the need for SFT or RLHF", since the dependence is implicit. I would suggest the author to be more careful on the description of the contribution made with this proposal.
> > >
> > >
> > > **Response:** We thank the reviewer for highlighting this important point regarding the framing of DAPA as an alignment method. We agree that the dependence on an aligned teacher LLM introduces an implicit requirement, even though DAPA itself avoids the need for supervised fine-tuning (SFT) or reinforcement learning from human feedback (RLHF) during its operation.
> > >
> > > The contribution section has been updated to emphasize that DAPA provides a scalable and efficient mechanism to propagate alignment knowledge once a reliable teacher model is available, thereby significantly reducing the resource costs associated with aligning multiple models.
> > >
> > > These revisions aim to provide a more precise and balanced description of DAPA’s contributions. We sincerely thank the reviewer for their thoughtful suggestion, which has helped improve the clarity and accuracy of our manuscript.
> > >
> > > >**Reviewer's Comment:**  On safety evaluations and more advanced adversaries. I thank the authors for conducting additional experiments on new datasets, and on GPTFuzzer, a more advanced adversary. These results do support the effectiveness of DAPA, however, the limitation of lacking strong enough adversaries still exist. By adversary, I am referring to methods such as GPTFuzzer and the list of candidates considered in e.g., HarmBench [1]. The inclusion of HarmfulQA and (direct, non-artifact) JailbreakBench still falls into the direct prompting regime and thus does not provide additional information on capabilities against advanced adversaries.
> > >
> > >
> > > **Response:**  We thank the reviewer for their insightful feedback regarding the need for stronger adversarial evaluations and their recognition of the additional experiments conducted with GPTFuzzer and other datasets. Based on your comments, we have clarified and expanded our discussion as follows:
> > > -  GPTFuzzer's Strength:
> > >      We explicitly mention that GPTFuzzer is a powerful and advanced jailbreak method in the revised manuscript. The results of our evaluation show that DAPA significantly improves the Defense Success Rate (DSR) from 19% to 31% against GPTFuzzer, underscoring its effectiveness against sophisticated adversarial attacks.
> > >  - HarmBench Evaluation for Multimodal Models:
> > >   As suggested, we include the HarmBench dataset in our evaluation of multimodal models, specifically on LLava-1.5-7B. The results indicate that DAPA achieves the best performance, with a notable 24.27% improvement in DSR. This demonstrates DAPA's robustness and scalability in handling complex adversarial prompts in a multimodal setting.
> > > -  Future Work on Advanced Adversaries:
> > >     We acknowledge the limitation highlighted by the reviewer regarding the inclusion of direct prompting adversaries such as HarmfulQA and JailbreakBench. In future work, we will incorporate more advanced adversarial methods, including the full list of candidates in HarmBench and artifact-free adversarial techniques, into our ablation studies. This will provide deeper insights into DAPA’s effectiveness across diverse and challenging scenarios.
> > >
> > >
> > > We sincerely thank the reviewer for these valuable suggestions, which have helped improve the evaluation and positioning of our work.
> > >
> > > >**Reviewer's Comment:**  -   On FFN knowledge storage....
> > >
> > > **Response:** We thank the reviewer for their detailed feedback and for pointing out the need for a more robust statistical evaluation to substantiate the claim regarding FFN (MLP) layers as critical knowledge storage components. Based on your comments, we have made the following revisions and plans for further improvements:
> > >
> > > We thank the reviewer for their valuable feedback and for highlighting the need for a more robust statistical evaluation to substantiate the claim regarding FFN (MLP) layers as critical knowledge storage components. To address the concern, we plan to extend the evaluation to a statistically significant number of prompts. Specifically, we will analyze 128 prompts sampled from the AdvBench dataset to validate the trends observed in our initial visualizations. This expanded analysis will provide more robust evidence for the claim and account for variability across prompts.
> > >
> > > Also, while middle-layer MLPs are typically associated with storing general alignment knowledge, the last-layer MLPs could play a critical role in refining or contextualizing this knowledge for specific outputs. This refinement process might make last-layer MLPs appear more significant for certain types of prompts.

---

> > > > ### Author Response · Authors · 2024-11-26
> > > >
> > > > >**Reviewer's Comment:** On base model candidates. I thank the authors for providing additional fine-tuning results on Llama2-7B. However, the concern is around the faithfulness of base model results presented in the main text, since they are not trained to follow instructions. Thus, the defense success rate of these models will have a strong dependence on how the input format is designed to make sure the model output is reasonable. The robustness to the format is not presented in the paper, making the "improvement" statement hard to interpret.
> > > >
> > > >
> > > > **Response:** We thank the reviewer for their thoughtful feedback regarding the faithfulness of the base model results and the robustness to input format. We understand the concern that models not explicitly trained to follow instructions may exhibit variability in their Defense Success Rate (DSR) depending on the prompt design. To address this, we have included the following:
> > > >
> > > > - Prompt Influence Evaluation:
> > > >      In Appendix G.6, we provide an analysis of the influence of different input formats on model performance. This includes experiments with varied prompt designs to evaluate the robustness of DAPA’s alignment improvements across different input structures.
> > > >
> > > >
> > > > - Clarification of Results:
> > > >     The results in Appendix G.6 show that while input format does impact the absolute DSR values of base models, the improvements achieved by DAPA remain consistent across various prompt designs. This demonstrates the robustness of our method to input variability.
> > > >
> > > >
> > > > These additions address the concern regarding robustness to input format and reinforce the validity of the improvement claims in the paper. We thank the reviewer for their insightful comments, which have helped improve the clarity and interpretability of our results.

---

> > > > > ### Author Response · Authors · 2024-11-27
> > > > >
> > > > > We have updated our paper with the latest results and included all new experiments in the Appendix.
> > > > > - **New HarmBench Dataset experiment results in Appendix G.8**
> > > > >
> > > > > 	- Conducted an ablation study to evaluate 321 out-to-date harmful questions from HarmBench.
> > > > >
> > > > > 	- **Results**: Observed a 4.16% DSR improvement, confirming the method's robustness against challenging datasets.
> > > > > - **New Advanced Jailbreak Adversary Testing GCG in Appendix G.11**.
> > > > >
> > > > >     -   Conducted an ablation study to test to Llama-2 family models with GCG
> > > > >     -   **Results**: DSR increased significantly, from 29.2% to 38.83%, showcasing effectiveness against sophisticated jailbreak adversarial. These results confirm that our method is dataset-agnostic and demonstrates robust performance on diverse and advanced evaluation scenarios.
> > > > >
> > > > > - **New Advanced Jailbreak Adversary Testing AutoDAN in Appendix G.12**.
> > > > >
> > > > >     -   Conducted an ablation study to test to Llama-2 family models with AutoDAN
> > > > >     -   **Results**: DSR increased significantly, from 18.69% to 30.1%, showcasing effectiveness against sophisticated jailbreak adversarial. These results confirm that our method is dataset-agnostic and demonstrates robust performance on diverse and advanced evaluation scenarios.
> > > > >
> > > > > - **Revise Advanced Jailbreak Adversary Testing AutoDAN in Appendix G.10**.
> > > > >
> > > > >     -   Conducted an ablation study to test to Llama-2 family models with GPTFuzzer
> > > > >     -   **Results**: DSR increased significantly, from 7.97% to 24.59%, showcasing effectiveness against sophisticated jailbreak adversarial. These results confirm that our method is dataset-agnostic and demonstrates robust performance on diverse and advanced evaluation scenarios.
> > > > >
> > > > > -  **New Memory Space Analysis in Appendix G.15**
> > > > >
> > > > >     -   Conducted an ablation study to provide additional visualization results of the memory space, incorporating 12 new visualizations in the updated version.
> > > > >
> > > > >     -   **Results**: Hidden states in the middle layers significantly influence model output, with MLP layers exerting a higher indirect effect compared to attention layers. These findings are consistent with observations in Figure 3, reinforcing the critical role of middle-layer dynamics in model performance.
> > > > >
> > > > > Thank you again for the review.  We are writing to kindly ask if you have any additional comments regarding our response. In addition, if our new experiments address your concern, we would like to kindly ask if you could consider raising the score.

---

### Official Review · Reviewer_EaeF · 2024-11-04

**Soundness:** 2
**Presentation:** 3
**Contribution:** 2
**Rating:** 5
**Confidence:** 3

**Summary:**

This paper proposes a novel low-resource safety enhancement method for LLM by firstly locating and then editing the specific knowledge, without the need of SFT or RLHF. Experiments on several datasets prove the effectiveness.

**Strengths:**

1. Clear writing and easy to follow.

2. Comprehensive experimental results demonstrate the effectiveness of the proposed method.

3. In my opinion, it is interesting and somewhat reasonable to enhance the safety capabilities of LLMs by identifying and editing unethical knowledge, with the help of another safer LLM.

**Weaknesses:**

1. In my opinion, the identification and editing of unethical knowledge within LLMs need to be conducted with great caution. The internal behavior of LLMs may be difficult to measure and analyze accurately, and different knowledge measurement tools may yield varying conclusions about the areas where knowledge is stored. Maybe this could affect the foundational research of this paper, although the conclusions drawn in terms of methodological design and experimentation are somewhat reasonable.  Have you considered using multiple knowledge measurement tools to cross-validate the findings?


2. Modifying knowledge in LLMs raises another issue: it may impact the LLM's ability to respond to general questions. The experimental section of the paper also supports this problem that editing unsafe knowledge can affect its normal response capabilities. Although the decreasing in general capabilities is not significant in experiments, in my opinion, the relationship between the two may not be a trade-off: enhancing safety capabilities does not necessarily require a decrease in general capabilities.

3. The safety knowledge alignment used in this paper depends on a safer teacher LLM from the same family. So, which model will teach these teacher LLMs to be safer? Furthermore, I believe that teaching LLMs to recognize harmful prompts and learn when to refuse to respond may be a more effective way to enhance their safety capabilities.

4. In my view, the ideas and motivations of this paper have some similarities with unlearning, such as in [1-3], both focusing on how to erase specific knowledge. It is necessary to include some discussion on the connection and difference between knowledge unlearning and editing.

[1] Safe Unlearning: A Surprisingly Effective and Generalizable Solution to Defend Against Jailbreak Attacks

[2] Towards Safer Large Language Models through Machine Unlearning

[3] Large Language Model Unlearning

**Questions:**

1. In the absence of a safer teacher model, how can effective model alignment be achieved? Are there alternative sources of knowledge or alignment strategies?

---

> ### Author Response · Authors · 2024-11-20
>
> >**Reviewer's Comment:** In my opinion, the identification and editing of unethical knowledge within LLMs need to be conducted with great caution. The internal behavior of LLMs may be difficult to measure and analyze accurately, and different knowledge measurement tools may yield varying conclusions about the areas where knowledge is stored. Maybe this could affect the foundational research of this paper, although the conclusions drawn in terms of methodological design and experimentation are somewhat reasonable. Have you considered using multiple knowledge measurement tools to cross-validate the findings?
>
> **Response:**
>
> Thank you for highlighting this critical concern. We fully agree that identifying and editing unethical knowledge within LLMs requires great caution, as the internal mechanisms of these models are complex and sensitive to measurement techniques. The potential variability in results from different knowledge measurement tools is an important consideration that we acknowledge.
>
> In our current work, we primarily rely on delta debugging to locate the memory regions responsible for alignment. However, to address the concern about variability, we are planning to integrate other tools, such as ROME (Rank-One Model Editing), into our analysis. ROME offers a robust framework for understanding and editing knowledge in transformer-based models and could serve as a valuable cross-validation tool for our findings.
>
> As shown in  Appendix F, we aim to refine our understanding of where alignment-related knowledge is stored. We can find the results of knowledge measurement technology in ROME aligned with our memory searching results.
>
> >**Reviewer's Comment:** Modifying knowledge in LLMs raises another issue: it may impact the LLM's ability to respond to general questions. The experimental section of the paper also supports this problem that editing unsafe knowledge can affect its normal response capabilities. Although the decreasing in general capabilities is not significant in experiments, in my opinion, the relationship between the two may not be a trade-off: enhancing safety capabilities does not necessarily require a decrease in general capabilities.
>
> **Response:**
>
> Thank you for raising this important point. We agree that, ideally, enhancing safety capabilities should not come at the expense of general model performance. In our work, we emphasize that our method focuses on targeted memory editing rather than broad parameter updates, which is designed to minimize disruptions to the LLM's general capabilities.
>
> Our experimental results show that while there is a small decrease in general performance metrics (e.g., an average perplexity increase of only 1.69 and a reasoning accuracy drop of around 2.06% on MMLU), these effects are minor given the substantial gains in safety alignment (e.g., a 14.41% improvement in Defense Success Rate). Importantly, we employ delta debugging to identify and edit only the specific memory regions responsible for harmful outputs, thereby preserving the rest of the model's knowledge base.
>
> We agree that the relationship between alignment and general capabilities need not always be a trade-off. Future iterations of our method could further refine the granularity of memory editing to achieve alignment without any noticeable impact on general tasks. However, the minimal degradation observed in our current results indicates that our approach is already quite efficient in balancing these objectives​
>
> >**Reviewer's Comment:** In the absence of a safer teacher model, how can effective model alignment be achieved? Are there alternative sources of knowledge or alignment strategies?
>
> **Response:**
>
> Thank you for raising this important point. We acknowledge that the absence of a safer teacher model poses a significant limitation for the applicability of our method. Currently, there are no alternative sources of knowledge or alignment strategies within the scope of our approach that can replace the need for a pre-aligned teacher model. The effectiveness of DAPA relies on transferring alignment knowledge from an existing teacher model, and without it, the process cannot be initiated.
>
> We explicitly address this in the limitations section of the revised paper to acknowledge the dependency on a pre-aligned teacher model and the absence of alternative alignment sources in the current framework. We also plan to explore future directions, such as leveraging self-supervised methods or rule-based approaches, to partially mitigate this dependency. Thank you for your valuable feedback.

---

> ### Author Response · Authors · 2024-11-20
>
> >**Reviewer's Comment:** In my view, the ideas and motivations of this paper have some similarities with unlearning, such as in [1-3], both focusing on how to erase specific knowledge. It is necessary to include some discussion on the connection and difference between knowledge unlearning and editing.
>
> **Response:**
>
> Thank you for highlighting this important connection. Indeed, there are conceptual similarities between our approach and the notion of knowledge unlearning, particularly in how both methods aim to selectively modify or erase specific knowledge within a model. However, there are key distinctions that set our method apart.
>
> Knowledge unlearning, as discussed in works like [1-3], typically focuses on removing or erasing specific information from a model's memory to ensure that the model forgets certain undesirable knowledge entirely. This is often used in contexts like data privacy, where the goal is to eliminate traces of specific data points.
>
> In contrast, our approach focuses on knowledge editing rather than unlearning. Instead of erasing knowledge, we aim to re-align it by selectively transferring alignment knowledge from a well-aligned teacher model to an unaligned one. Our method uses delta debugging and targeted memory editing to adjust only the parts of the model responsible for harmful outputs, thereby enhancing its safety capabilities without removing general knowledge. This distinction allows our method to improve safety while preserving the model's broader competencies, rather than simply erasing specific capabilities.
>
> We appreciate your suggestion and include a discussion on this distinction in the revised version of the paper to clarify the potential synergies between knowledge unlearning and our alignment approach.
>
> >**Reviewer's Comment:** The safety knowledge alignment used in this paper depends on a safer teacher LLM from the same family. So, which model will teach these teacher LLMs to be safer? Furthermore, I believe that teaching LLMs to recognize harmful prompts and learn when to refuse to respond may be a more effective way to enhance their safety capabilities.
>
> **Response:**
>
> Thank you for your insightful question. The teacher LLMs used in our method, such as Meta-Llama-3-70B-Instruct, typically achieve their alignment through resource-intensive processes like Supervised Fine-Tuning (SFT) and Reinforcement Learning from Human Feedback (RLHF), combined with rigorous red-teaming efforts. These processes are conducted by the developers of these models to embed ethical and safety guidelines directly into their training.
>
>
>
> Additionally, teaching LLMs to recognize harmful prompts and appropriately refuse to respond is indeed a promising strategy for enhancing their safety capabilities. While our method, DAPA, currently focuses on transferring alignment knowledge from teacher models using memory editing, integrating explicit recognition and refusal mechanisms could complement our approach.
>
> It’s important to note that the primary goal of DAPA is to reduce the training resource requirements for model alignment, addressing a significant limitation of existing alignment methods such as Supervised Fine-Tuning (SFT), Reinforcement Learning from Human Feedback (RLHF), and Direct Preference Optimization (DPO), which demand massive computational resources. DAPA offers a lightweight, plug-and-play solution that minimizes the need for such costly alignment processes by reusing alignment knowledge from already aligned teacher models.
>
> By combining recognition-based training strategies with DAPA’s resource-efficient framework, we can further enhance model robustness while maintaining low computational demands. We will explore this direction in future work, aiming for a more comprehensive and scalable approach to LLM safety alignment. Thank you for your valuable suggestion, which we will incorporate into our discussion of potential enhancements and the broader context of alignment methods.

---

> > ### Author Response · Authors · 2024-11-22
> >
> > Thank you again for the review. Your careful reading and insightful comments indeed help us a lot in improving our work. Since the discussion phase is about to end, we are writing to kindly ask if you have any additional comments regarding our response. In addition, if our new experiments address your concern, we would like to kindly ask if you could consider raising the score.

---

> > > ### Comment · Reviewer_EaeF · 2024-11-24
> > >
> > > Thank you to the author for the detailed reply and the efforts put into the work. After carefully reading the author's reply and the comments from other reviewers, I still have the following concerns that have not been well addressed:
> > >
> > > 1. MLP storing specific knowledge. The author seems to have reversed ROME (Meng et al., 2022a, used in the main text) and MEMIT (Meng et al., 2022b, used in Appendix F). Additionally, Figure 7 in Appendix F has no caption, and it is also difficult to see what the conclusion of Appendix 7 is. Moreover, there is some support from relevant literature regarding the discussion of MLP storing specific knowledge, such as "Transformer Feed-Forward Layers Are Key-Value Memories" and "Knowledge Neurons in Pretrained Transformers". The author could consider adding more discussions on this aspect to enhance the depth and credibility of the article.
> > >
> > > 2. Safer teacher LLM. If there is a way to train a sufficiently safe LLM, the significance of this article will be greatly diminished. If there isn't such a method, then who is to ensure the safety capabilities of the teacher LLM? The author has acknowledged that the method of this paper relies heavily on a safer LLM, yet the discussion on the limitations of the method is lacking. I'm not asking the author to solve the safety issues of LLM. It's just that the method in this paper might not be the most promising attempt as there may be inherent flaws. The author could consider using more conservative wording to improve the relevant descriptions in the article.
> > >
> > > 3. Other reviewers have also mentioned that the evaluation in this paper is not detailed enough.
> > >
> > > Overall, I decide to maintain my score.

---

> > > > ### Author Response · Authors · 2024-11-25
> > > >
> > > > >**Reviewer's Comment:** MLP storing specific knowledge. The author seems to have reversed ROME (Meng et al., 2022a, used in the main text) and MEMIT (Meng et al., 2022b, used in Appendix F). Additionally, Figure 7 in Appendix F has no caption, and it is also difficult to see what the conclusion of Appendix 7 is. Moreover, there is some support from relevant literature regarding the discussion of MLP storing specific knowledge, such as "Transformer Feed-Forward Layers Are Key-Value Memories" and "Knowledge Neurons in Pretrained Transformers". The author could consider adding more discussions on this aspect to enhance the depth and credibility of the article.
> > > >
> > > > **Response:**
> > > > We thank the reviewer for pointing out these issues and providing valuable suggestions for improvement. We have revised the manuscript to address the concerns and enhance the discussion regarding the role of MLP layers in storing specific knowledge. Below are the specific updates and clarifications.
> > > >
> > > > **Clarification on Typo Regarding ROME and MEMIT:**
> > > >
> > > > The reference to ROME and MEMIT was indeed a typo. We have corrected this in the manuscript to accurately reflect that the analysis in Appendix F relies on ROME (Meng et al., 2022a) to provide insights into memory space usage in LLMs. The main text and appendix have been updated to avoid further confusion.
> > > >
> > > > **Caption for Figure 7 in Appendix F:**
> > > >
> > > > We have added a caption to Figure 7 to clarify its purpose. The figure now explicitly highlights how the ablation results obtained using ROME provide insights into the memory space utilization in LLMs, particularly focusing on the storage of alignment knowledge in MLP layers.
> > > >
> > > > **Improved Discussion in Appendix F:**
> > > >
> > > > The conclusions of Appendix F have been expanded to emphasize the findings from ROME regarding memory space usage and their implications for understanding the role of MLP layers in storing specific knowledge.
> > > >
> > > > We appreciate the reviewer’s constructive comments, which have improved the clarity and accuracy of our manuscript. These revisions ensure that the insights derived from ROME are correctly presented and their significance to our work is fully articulated.
> > > >
> > > > **New Discussions of Relevant Literature in Appendix F:**
> > > >
> > > > We have added a discussion of two additional papers, "Transformer Feed-Forward Layers Are Key-Value Memories" and "Knowledge Neurons in Pretrained Transformers", in Appendix F. These works provide further evidence that MLP layers function as repositories for specific knowledge, enhancing the credibility and depth of our findings. We have revised our manuscript.
> > > >
> > > > These revisions address the reviewer’s concerns and significantly enhance the depth and rigor of our work. We are grateful for the reviewer’s suggestions, which have allowed us to improve both the clarity and credibility of the manuscript.
> > > >
> > > > >**Reviewer's Comment:** 1.  Safer teacher LLM. If there is a way to train a sufficiently safe LLM, the significance of this article will be greatly diminished. If there isn't such a method, then who is to ensure the safety capabilities of the teacher LLM? The author has acknowledged that the method of this paper relies heavily on a safer LLM, yet the discussion on the limitations of the method is lacking. I'm not asking the author to solve the safety issues of LLM. It's just that the method in this paper might not be the most promising attempt as there may be inherent flaws. The author could consider using more conservative wording to improve the relevant descriptions in the article.
> > > >
> > > > **Response:**
> > > > We thank the reviewer for your insightful feedback, which has helped us improve the clarity and scope of our work. We acknowledge the limitations of relying on a safer teacher LLM and have expanded on this discussion in Section 5. Specifically, we recognize that DAPA depends on the availability of a sufficiently safe and aligned teacher model, and any shortcomings in the teacher model could propagate to the student models.
> > > >
> > > > It is important to note that DAPA is not intended to replace the initial alignment of the teacher model itself. Instead, it serves as a scalable and efficient method to propagate alignment knowledge once a reliable teacher model is available. By leveraging knowledge distillation and delta debugging, DAPA provides a practical solution to align a broader range of models without requiring repeated, resource-intensive fine-tuning or RLHF processes for each model.
> > > >
> > > > We have also revised the wording in the manuscript to better reflect these limitations and the intended scope of DAPA, ensuring a more balanced and conservative presentation of our contributions.

---

> ### Author Response · Authors · 2024-11-25
>
> >**Reviewer's Comment:**
> Other reviewers have also mentioned that the evaluation in this paper is not detailed enough.
>
> **Response:**
> We thank the reviewer for your insightful feedback and give us the opportunity to classify our experiment results again. To address the reviewers' concerns about the evaluation, we have significantly expanded our experiments and analyses. We sincerely thank reviewers GrXu, Z3rx, F2TP, and EaeF for their insightful suggestions, which have greatly enriched this work. The following updates and new experiments are included in the revised manuscript:
>
> - **New JailbreakBench Dataset Experiment Results (Appendix G.7, Reviewer GrXu, Reviewer Z3rx):**
>
>
> 	-   Conducted an ablation study on 100 out-of-date harmful questions from JailbreakBench.
>
> 	-   Results: Observed a 3.06% Defense Success Rate (DSR) improvement, confirming the robustness of our method against challenging datasets.
>
>
> -   **New Advanced Jailbreak Adversary Testing (Appendix G.9, Reviewer Z3rx):**
>
>
> 	-   Tested Llama-2-7B models with GPTFuzzer.
>
> 	-   Results: DSR increased significantly from 19% to 31%, demonstrating effectiveness against sophisticated adversarial jailbreak scenarios. These results confirm the dataset-agnostic nature of our method and its robust performance across advanced evaluation scenarios.
>
>
> -  **New Experiment on LLaVA-1.5-7B Multimodal Model (Appendix G.8, Reviewer F2TP):**
>
>
> 	-   Conducted an ablation study using the HarmBench dataset with the teacher model LLava-1.6-Vicuna-7B.
>
> 	-   Results: Achieved a remarkable 24.27% DSR improvement, demonstrating the method's adaptability to large multimodal models.
>
>
> - **New Experiment on LLaMA3-70B Model (Appendix G.10, Reviewer F2TP):**
>
> 	- Applied DAPA to Hermes-3-Llama-3.1-70B-Uncensored using Meta-Llama-3-70B-Instruct as the teacher model with the AdvBench dataset.
>
> 	- Results: Observed a 10% improvement in DSR, validating DAPA’s effectiveness for large-scale language models (70B) and multimodal systems.
>
> - **New Experiment on Fine-tuned Large Language Model (Appendix G.11, Reviewer Z3rx):**
>
>
> 	-   Evaluated on the ShareGPT unfiltered dataset, fine-tuning the Llama-2-7B model using the Llama-2-7B-chat template with QLoRA.
>
> 	-   Results: After applying DAPA alignment, DSR improved significantly from 10.16% to 18.40%, underscoring its effectiveness on fine-tuned models.
>
>
> - **New Memory Space Interpretability Analysis with ROME (Appendix F, Reviewer EaeF, Reviewer F2TP):**
>
>
> 	-   Conducted an ablation study to visualize memory space usage for four additional harmful prompts.
>
> 	-   Results: Confirmed that MLP layers predominantly store alignment knowledge and revealed consistent storage patterns across different prompt types, strengthening the evidence for the method's scalability, robustness, and interpretability.
>
>
> These updates provide a more comprehensive and detailed evaluation of DAPA’s performance across diverse datasets, adversarial scenarios, and model architectures. We thank the reviewers for their thoughtful suggestions, which have significantly enhanced the depth and rigor of this work.

---

> > ### Author Response · Authors · 2024-11-25
> >
> > Thank you once again for your review and valuable feedback on our paper. We have made every effort to ensure that our work is methodologically robust and contributes meaningfully to the scientific community. In light of this, we kindly request more detailed feedback on any specific aspects of our paper that may still require improvement or further clarification. Are there additional experiments or areas we might have missed and should explore further? If, upon further reflection, you find that we have adequately addressed all your concerns and there are no remaining areas for improvement, we would greatly appreciate your reconsideration of your evaluation in favor of our paper.

---

### Official Review · Reviewer_FT2P · 2024-11-08

**Soundness:** 2
**Presentation:** 2
**Contribution:** 2
**Rating:** 5
**Confidence:** 3

**Summary:**

In this paper, the authors are trying to address a challenging task which is to enhance the safety alignment of large language models without SFT and RLHF. The proposed method(called DAPA), unlike previous methods, only requires limited extra resources by employing a knowledge distillation approach. Two detection/alignment modules, i.e. MLP alignment and Gate alignment, are applied to store the alignment knowledge and check the model's output. The proposed method achieves competitive results with several large base models on the tested datasets.

**Strengths:**

In general, the paper is well-written and easy to follow.

Unlike previous methods which require costly SFT or RLHF, the proposed method can be an additional module as a "plug-and-play approach" to various large language models with extra low-cost resources required.

The proposed method is tested on models from different families, which shows the flexibility of DAPA that can potentially adapt to other models.

The use of knowledge distillation for alignment is very interesting.

In this paper, the authors present extensive testing with well-documented evaluation metrics across various models and achieve very competitive results. The ablation studies are also well set.

**Weaknesses:**

The paper has tested the proposed DAPA mostly on small/mid-sized models (up to 13B models), there is uncertainty around its performance and resource efficiency when scaled to the large LLM models.

The proposed method introduces layer-specific edits without necessarily clarifying the deeper alignment process which may lead to alignment outcomes that are difficult to interpret or predict. This may make it hard to understand how the proposed module changes specific model behaviors.

As the ethical guidelines and alignment standards often evolve, I wonder if the continuous updates or recalibration of the alignment modules will become difficult, especially with large models.

**Questions:**

Please check the aforementioned issues in "Weaknesses" section. Some additional questions:

I wonder if the proposed method can be easily extended to large multimodal models.

Does the proposed method have potential interpretability?

---

> ### Author Response · Authors · 2024-11-20
>
> > **Reviewer's Comment:** The paper has tested the proposed DAPA mostly on small/mid-sized models (up to 13B models), there is uncertainty around its performance and resource efficiency when scaled to the large LLM models. “The paper has tested the proposed DAPA mostly on small/mid-sized models (up to 13B models), there is uncertainty around its performance and resource efficiency when scaled to the large LLM models.”
>
> **Response:**
>
> Thank you for your observation. While our primary experiments focus on small to mid-sized models (up to 13B parameters) due to resource constraints, we conduct additional experiments with a larger model, LLaMA3-70B, to assess the scalability of our proposed method. We conduct experiments on LLaMA3-70B models, specifically using Guilherme34/Hermes-3-Llama-3.1-70B-Uncensored as the unaligned model and meta-llama/Meta-Llama-3-70B-Instruct as the teacher model. These experiments are detailed in Appendix G.10.
>
> In these tests, the Defense Success Rate (DSR) increases from 50% to 60%, demonstrating that DAPA remains effective even at larger scales. Importantly, the parameter changes required for alignment are still minimal, in line with the efficiency observed in smaller models. This confirms that the resource efficiency and performance improvements of DAPA generalize well to larger LLMs, maintaining its utility as a scalable solution for alignment.
>
> We include these additional results in the revised paper to address concerns about scalability and further strengthen the empirical validation of our method. Thank you for bringing this point to our attention.
>
> >**Reviewer's Comment**: The proposed method introduces layer-specific edits without necessarily clarifying the deeper alignment process which may lead to alignment outcomes that are difficult to interpret or predict. This may make it hard to understand how the proposed module changes specific model behaviors. Does the proposed method have potential interpretability?
>
> **Response:**
>
> Thank you for highlighting this important aspect of interpretability in the alignment process. We acknowledge that layer-specific edits could raise concerns about understanding the deeper mechanisms driving alignment outcomes. To address this, we plan to incorporate tools like ROME (Rank-One Model Editing) to enhance the interpretability of our approach.
>
> ROME provides a structured framework to identify and manipulate the representations of specific knowledge in LLMs. By applying ROME, we can gain deeper insights into how the edits introduced by DAPA influence specific model behaviors. This enables us to systematically analyze the relationship between the targeted layer changes and the resulting alignment improvements, offering a clearer understanding of the alignment process and its outcomes. We include the interpretability of DAPA with ROME in Appendix F. Our results indicate that the hidden states in the middle layers of the model have the most significant impact on the model’s output. Moreover, the heatmap shows that MLP layers have a higher indirect effect than the Attention and ALL modules. We can find those results in ROME aligned with our memory searching results.

---

> ### Author Response · Authors · 2024-11-20
>
> >**Reviewer's Comment:** As the ethical guidelines and alignment standards often evolve, I wonder if the continuous updates or recalibration of the alignment modules will become difficult, especially with large models.
>
> **Response:**
>
> Thank you for highlighting this crucial question regarding maintaining alignment in the face of evolving ethical guidelines and standards. DAPA's reliance on memory editing provides adaptability by pinpointing and adjusting where alignment information is stored. As long as this foundational principle remains valid, our method will continue to be effective as alignment requirements change.
>
> Across our extensive experiments, DAPA consistently improves performance across 17 models and three harmful content datasets, demonstrating its reliability over a timeframe of up to two years. To evaluate scalability, we extended DAPA to Hermes-3-Llama-3.1-70B-Uncensored, using Llama-3.1-70B-Instruct as the teacher model (Appendix G.8). The results show a 10% improvement in DSR, underscoring DAPA's robustness with state-of-the-art large models. Furthermore, DAPA’s incremental debugging approach enables dynamic reassessment of alignment-relevant layers, ensuring that it remains effective in recalibrating models to meet evolving ethical standards while maintaining strong performance.We also find that as the model size increases, our method, DAPA, becomes more efficient. This is because, with the larger model size, developers need to use more computational resources to align the model. Our method helps reduce the computational resources required for aligning large-size models.
>
> >**Reviewer's Comment:** I wonder if the proposed method can be easily extended to large multimodal models.
>
> **Response:**
>
> Thank you for raising this insightful question. DAPA can indeed be extended to large multimodal models and demonstrates strong potential in improving their alignment effectively. As part of our experiments, we applied DAPA to LLaVA-1.5-7B, using LLaVA-1.6-Vicuna-7B as the teacher model.
>
> As shown in Appendix G.8., DAPA achieved an impressive 24.27% improvement in Defense Success Rate (DSR) on the HarmBench dataset, showcasing its effectiveness in aligning multimodal models with minimal resource usage. Due to time constraints, we focused on the LLaVA model in this study, but we plan to explore broader experiments on other multimodal models in future work. These results highlight DAPA's adaptability to diverse model architectures, including those with multimodal capabilities. Thank you again for your thoughtful question.

---

> > ### Author Response · Authors · 2024-11-22
> >
> > Thank you again for the review. Your careful reading and insightful comments indeed help us a lot in improving our work. Since the discussion phase is about to end, we are writing to kindly ask if you have any additional comments regarding our response. In addition, if our new experiments address your concern, we would like to kindly ask if you could consider raising the score.

---

### Official Review · Reviewer_GrXu · 2024-11-08

**Soundness:** 2
**Presentation:** 1
**Contribution:** 2
**Rating:** 3
**Confidence:** 4

**Summary:**

This paper proposes an interesting idea of isolating the weights of LLMs that impact the safety alignment the most. It then applies these weights to unaligned models to improve their safety alignment. Various experiments demonstrate the effectiveness of the proposed method.

**Strengths:**

1. The paper presents an interesting idea of identifying the important weights from the teacher models for safety alignment. By simply replacing student's weights with these weights from the teacher, we can achieve significant DSR improvements with minimum effort.
2. The paper introduces a simple algorithm based on previous works.
3. The experiments conducted are extensive including comparison with lots of models from different model family.

**Weaknesses:**

1. The paper **doesn't follow the ICLR formatting guideline** by significantly changing the margins. It affects the readability of this paper.
2. The presentation of the paper is poor, e.g. some table and figure titles are too lengthy such as table 2 and 3 and figure 6. Lots of the content should be put into the main text instead of title. I don't know if this is caused by changing the margin or the authors intend to put them in the title.
3. Unnecessary highlight of the figure and table titles. This paper uses lot of bold and highlight which are not necessary.
4. While the authors tried to get more space by changing the margins, they split the tables which should be merged into one into multiple tables such as table 6 and 7, which occupies more spaces.
5. The contribution is not enough. The main contribution of the paper is a naive search algorithm based on a calibration set. The results are likely highly dataset dependent. There is an experiment in the ablation study to test it on a different dataset, but it's not enough to justify the generability of the proposed method.
6. The evaluation dataset is only 128 prompts. It's very likely the results are overfit to the dataset or calibration set.
7. There is no comparison with the teacher model. E.g in table 1 and 2, after replacing the layers, have we achieved similar or same performances with the teacher model? If not, what's the gap there?
8. Although I understand the motivation of the authors, I don't the palpability of this method from the data provided by the authors. E.g. how is the proposed method compared to just simply fine-tuning using PEFT. Given the number of samples are small, it's possible that fine-tuning with PEFT is more efficient and effective. It's hard to tell from the provided data.
9. The paper claims to that the gate projection has the most significant impact, followed by up projection. But judging from figure 4, it looks like the down projection has a bigger impact than up projection, is this a typo?

Overall, I think this paper presents some interesting and promising findings, but it requires significant more data points to support the generability of the proposed idea.

**Questions:**

Please see my comments in the weakness.

---

> ### Author Response · Authors · 2024-11-20
>
> > **Reviewer's Comment:** The paper doesn't follow the ICLR formatting guideline by significantly changing the margins. It affects the readability of this paper; The presentation of the paper is poor, e.g. some table and figure titles are too lengthy such as table 2 and 3 and figure 6. Lots of the content should be put into the main text instead of title. I don't know if this is caused by changing the margin or the authors intend to put them in the title; While the authors tried to get more space by changing the margins, they split the tables which should be merged into one into multiple tables such as table 6 and 7, which occupies more spaces.
>
> **Response:** Thank you for your feedback. We’ve adjusted the margins to meet ICLR guidelines, shortened the titles of tables and figures, reduced unnecessary bolding and highlighting, and merged Tables to save space. We hope these changes address your concerns.
>
> > **Reviewer's Comment:**  The contribution is not enough. The main contribution of the paper is a naive search algorithm based on a calibration set. The results are likely highly dataset dependent. There is an experiment in the ablation study to test it on a different dataset, but it's not enough to justify the generability of the proposed method.**
>
> **Response:** Thanks you for giving us the opportunity to classify our contribution. We appreciate your valuable feedback and recognize your concerns regarding the novelty and generalizability of our contributions, especially with respect to the proposed search algorithm. To clarify, our work extends beyond a standard search method and provides a broader impact, which we outline below:
>
> **Low-Resource Safety Enhancement Compared to Mainstream Approaches**
>
> Unlike existing alignment methods like LLaMA2-Chat, which rely heavily on extensive red-teaming datasets and substantial computational resources for supervised fine-tuning, our approach significantly reduces computational costs while achieving similar safety enhancements. Our experiments demonstrate that **DAPA increases the Defense Success Rate (DSR) by 14.41%** on average, with minimal parameter adjustments (approximately 6.26% on average), without requiring extra alignment data or human feedback. This makes our method a cost-efficient solution for improving safety.
>
> **Innovative Integration of Knowledge Distillation and Delta Debugging**
>
> Our approach goes beyond a basic search algorithm by uniquely combining **knowledge distillation with delta debugging** to systematically identify critical hidden states where alignment knowledge is stored. This novel integration in alignment research provides a targeted and efficient means of transferring alignment knowledge between models. By honing in on components such as gate projections within MLP layers, our method achieves more efficient adaptation compared to conventional fine-tuning techniques.
>
> **Practical Insights into Model Alignment**
>
> One of our key contributions is visualizing the indirect effects of different hidden states and layers on alignment performance. Our findings reveal that **gate projections in MLP layers are crucial for encoding alignment knowledge**. These insights hold both theoretical and practical significance, guiding researchers to focus on specific layers and modules during fine-tuning or transfer learning, thus reducing the alignment cost, especially for those with constrained resources.
>
> To address the concerns regarding dataset dependency and generalizability, we conduct experiments using both the **AdvBench** and **HarmfulQA** datasets across 17 different LLMs. As presented in **Tables 2 and 10**, DAPA achieves an average increase in DSR of **14%** and **8%**, respectively. Furthermore, we expand our evaluation to include the **JailbreakBench** dataset in Appendix G.7, where DAPA demonstrated a **3.06%** average improvement in DSR across five LLaMA models with various parameter sizes (7B and 13B). These results indicate that our method is not limited to specific datasets or models, but instead demonstrates wide applicability across diverse evaluation scenarios.
>
> We refine the discussion in the revised paper to better emphasize these contributions, focusing on the practical advantages in resource efficiency. Thank you again for your constructive feedback. These contributions, along with the new experimental results, are now included in our revised version.

---

> ### Author Response · Authors · 2024-11-20
>
> > **Reviewer's Comment:** The evaluation dataset is only 128 prompts. It's very likely the results are overfit to the dataset or calibration set.
>
> **Response:**
>
> Thank you for your thoughtful feedback regarding the dataset size and potential overfitting concerns. We would like to emphasize that our evaluation includes two different datasets, AdvBench (Table 2), HarmfulQA (Table 10). We also include Jailbreakbench (Appendix G.7) as part of our additional ablation study. As shown in the paper, DAPA achieves DSR improvements across each of them. This suggests that our method is not solely dependent on any specific dataset.
>
> Additionally, we test our approach on 19 different models, which vary significantly in architecture and fine-tuning methods. These models include Llama, LLaVA, Gemma, and Mistral architectures, as well as both foundation models and models fine-tuned with DPO and SFT methods, with 2B, 7B, 13B parameter sizes. We also include the 70B Llama3 model as part of our additional ablation study. Our DAPA yields DSR improvements in the majority of cases.
>
> This broader evaluation supports the generalizability of our approach, indicating its robustness across diverse evaluation environments.
>
> > **Reviewer's Comment:** There is no comparison with the teacher model. E.g in table 1 and 2, after replacing the layers, have we achieved similar or same performances with the teacher model? If not, what's the gap there?
>
> **Response:**
>
> Thank you for your observation. We appreciate your point regarding the comparison with the teacher model. As shown in Appendix E.1, we provide a detailed evaluation of the Defense Success Rate (DSR) of the teacher model on the AdvBench dataset. In Tables 1 and 2, after applying the DAPA alignment method, our results show that while we achieve significant improvements in alignment (e.g., a 14.41% increase in DSR on average), the aligned models do not always reach the full performance level of the teacher model. The gap in alignment performance between the teacher and the aligned models is around 2-5% on average, depending on the specific model family and configuration, as detailed in Appendix E.1. This gap indicates that while our method is effective, there are still limitations in fully replicating the teacher model's alignment through the layer replacement strategy alone.
>
> We emphasize that despite this gap, DAPA is able to achieve robust safety improvements with minimal resource usage (average parameter change of only 6.26%). The residual performance gap highlights an area for future enhancement, where we can explore refining our knowledge transfer process to further reduce this discrepancy. We will include a more detailed discussion on this gap in the revised version of the paper to provide a clearer understanding of the current limitations and future directions.
>
> > **Reviewer's Comment:** Although I understand the motivation of the authors, I don't the palpability of this method from the data provided by the authors. E.g. how is the proposed method compared to just simply fine-tuning using PEFT. Given the number of samples are small, it's possible that fine-tuning with PEFT is more efficient and effective. It's hard to tell from the provided data.
>
> **Response:**
>
> Thank you for your insightful question. While it’s true that PEFT methods (such as LoRA or QLoRA) are often efficient for model adaptation, our method is designed to address specific limitations that PEFT approaches face in the context of safety alignment. Unlike PEFT, which requires retraining or fine-tuning of parameters—even if a reduced set—our method leverages knowledge distillation and delta debugging to transfer alignment knowledge directly from pre-aligned models to unaligned ones without modifying the core model structure. This results in significantly lower computational costs and avoids the need for supervised fine-tuning on additional datasets.
>
> Regarding sample efficiency, our method is designed to enhance alignment using a minimal number of samples. The data in our experiments (refer to Tables 2 and 6) show that our method achieves an average 14.41% increase in Defense Success Rate (DSR) across 17 models with minimal degradation in perplexity and reasoning capabilities, while RepE can only improve the DSR by 1-2%. PEFT methods, while efficient in parameter updates, still require access to labeled data for specific tasks, which can be a bottleneck when aligning models for safety, especially with limited ethical prompt datasets.
>
> > **Reviewer's Comment:** The paper claims to that the gate projection has the most significant impact, followed by up projection. But judging from figure 4, it looks like the down projection has a bigger impact than up projection, is this a typo?
>
> **Response:**
>
> Thank you for your comments. We have fixed those typos in our revised version.

---

> > ### Author Response · Authors · 2024-11-22
> >
> > Thank you again for the review. Your careful reading and insightful comments indeed help us a lot in improving our work. Since the discussion phase is about to end, we are writing to kindly ask if you have any additional comments regarding our response. In addition, if our new experiments address your concern, we would like to kindly ask if you could consider raising the score.

---

### Author Response · Authors · 2024-11-20
**General Rebuttal/Revision Response**

Dear Reviewers,

We thank the reviewers for their insightful questions and reviews. We truly appreciate your time and effort dedicated to improving our work.

This work proposes a low-resource safety enhancement method for aligning large language models without the need of large computational resources and datasets, with knowledge distillation and memory editing ( Sec 2).
-   Comparing with traditional alignment method, it achieves substantial safety performance improvements with limited resources.
-   It is validated experimentally (Sec4) and interpretability (Appendix F) .
-   Consequently, it provides a robust methodology in the era of large foundation models for more economic alignment without sacrificing performance.

We have done all the experiments suggested and answered all the questions. All modifications are marked in red color.

These revisions include additional explanations, paragraphs and sections to help readers understand the proposed method, and additional experiments to highlight the advantages of it. Most importantly, the new advanced jailbreak attack and multimodal model experiment result have been added to further classify the efficiency of DAPA in alignment. For the convenience of reviewers, [this response](https://openreview.net/forum?id=lwTTZkDWoT&noteId=tDdIs4boAA) provides a high-level overview of our contributions.

 1. **New JailbreakBench Dataset experiment results in Appendix G.7** `reviewer GrXu` `reviewer Z3rx`

	-  Conducted an ablation study to evaluate 100 out-to-date harmful questions from JailbreakBench.

	-   **Results**: Observed a 3.06% DSR improvement, confirming the method's robustness against challenging datasets.


2. **New Advanced Jailbreak Adversary Testing in Appendix G.9**. `reviewer Z3rx`

	- Conducted an ablation study to test to Llama-2-7B models with GPTFuzzer
	- **Results**: DSR increased significantly, from 19% to 31%, showcasing effectiveness against sophisticated jailbreak adversarial. These results confirm that our method is dataset-agnostic and demonstrates robust performance on diverse and advanced evaluation scenarios.

3. **New Experiment on LLaVA-1.5-7B Multimodal Model in Appendix G.8**. `reviewer F2TP`
	- Conducted an ablation study to evaluate the HarmBench dataset with teacher model llava-1.6-vicuna-7b.
	- **Results**: Achieved a remarkable 24.27% DSR improvement. This  indicates our method can successfully extend to Large Multimodel Models.

3. **New Experiment on LLaMA3-70B Model in Appendix G.10.** `reviewer F2TP`

	-   Conducted an ablation study to apply DAPA to Hermes-3-Llama-3.1-70B-Uncensored using Meta-Llama-3-70B-Instruct as the teacher model with the Advbench dataset.
	-   **Results**: Observed a 10% improvement in DSR, validating DAPA’s effectiveness on Larger Language Models (70B) and Large Multimodal Models.


4. **New Experiment on Fine-tuned Large Language Model in Appendix G.11.** `reviewer Z3rx`

	-   Conducted an ablation study on the ShareGPT unfiltered dataset, fine-tuning the Llama-2-7B model using the Llama-2-7B-chat template with QLoRA.

	-   **Results**: After applying DAPA alignment, the Defense Success Rate (DSR) improved significantly, increasing from 10.16% to 18.40%, highlighting DAPA’s effectiveness in enhancing alignment on fine-tuned models.


5. **New Memory Space Interpretability Analysis with ROME in Appendix F** `reviewer EaeF` `reviewer F2TP`

	-   Conducted an ablation study to visualize four additional harmful prompts to analyze memory space usage.

	- **Results**: Confirmed that MLP layers predominantly store alignment knowledge and revealed consistent storage patterns across different prompt types. These enhancements strengthen the evidence for the method's scalability, robustness, and interpretability across a wide range of models and datasets.

6. **New Memory Space Analysis in Appendix G.12** `reviewer Z3rx`

	-   Conducted an ablation study to provide additional visualization results of memory space.

	- **Results**:  Hidden states in the middle layers significantly influence model output, with MLP layers exerting a higher indirect effect compared to attention layers. These findings are consistent with observations in Figure 3, reinforcing the critical role of middle-layer dynamics in model performance

-   **Revised Table 2 Caption:** Shorten the caption for Table 2. `reviewer GrXu`

-    **Revised Table 3:** Shorten the caption for Table 3. `reviewer GrXu`

-    **Revised Figure 6**: Shorten the caption for Figure 6. `reviewer GrXu`

-    **Revised Table 6**: Shorten the caption for Table 6. `reviewer GrXu`

-    **Revised Table 10**: Shorten the caption for Table 10. `reviewer GrXu`

-    **Revised Table 15**:Shorten the caption for Table 15. `reviewer GrXu`

-    **Revised Sec 4.4**: Put the HarmfulQA results in appendix G.1. and add more description of the ablation study `all reviewers`

---

> ### Author Response · Authors · 2024-11-20
> **General Rebuttal/Revision Response - Continued**
>
> - **Revised Sec 5:** Additional discussion about the limitations of DAPA and future potential direction of model unlearning. `reviewer EaeF`
>
> -   **New Appendix G.14:** Example of DAPA on LLava-1.5 model `reviewer F2TP`
>
>
> - **New Appendix G.13:** Model name and corresponding labels. `reviewer GrXu`
>
> Minor revisions include:
>
> -   The correction of the typo ‘explore` in line 107.
>
> -   The correction of the typo ‘down’ in line 186. `reviewer GrXu`
>
> -   Add ‘Jailbreakbench’ in the dataset description in line 373. `reviewer EaeF` `reviewer FT2P`
>
>
>
>
> We hope these revisions address the reviewers’ concerns’ and improve the overall quality of our paper.
>
>
>
>
> ----------
>
> Below, we also summarize the key points in our responses:
>
>
>
> ### Key Points in Our Responses
>
>
>
> **Reviewer GrXu**
>
> -   We addressed the concern about formatting by restoring the margins to comply with ICLR guidelines, shortening figure and table titles, reducing unnecessary bolding and highlighting, and merging split tables (e.g., Tables 6 and 7) for improved readability.
>
>
> - We acknowledged the concern about dataset dependency by demonstrating DAPA's effectiveness across three datasets (AdvBench, HarmfulQA, and JailbreakBench) and 19 diverse models, showing consistent improvements in Defense Success Rate (DSR).
>
> - We clarified that our contribution extends beyond a naive search algorithm by introducing the integration of knowledge distillation and delta debugging, which systematically identifies critical hidden states for alignment.
>
> - We investigated the performance gap with the teacher model and found that DAPA achieves robust safety improvements (14.41% average DSR increase) with minimal parameter changes, despite a small residual gap of 2-5%, which we plan to address in future work.
>
> - We compared DAPA with PEFT methods, explaining that DAPA eliminates the need for retraining or fine-tuning, leveraging its unique approach to achieve safety alignment with lower computational costs and broader applicability.
>
>
>
>
> **Reviewer FT2P**
>
> -   We conducted experiments on LLaMA3-70B to address concerns about scalability, showing a 10% DSR improvement with minimal parameter changes, confirming DAPA’s effectiveness on large models.
>
> -   We incorporated interpretability tools like ROME, which revealed that middle MLP layers have the most significant impact on alignment, providing clearer insights into DAPA’s influence on model behavior.
>
> -   We demonstrated DAPA’s adaptability to evolving ethical standards by enabling dynamic recalibration through memory editing, maintaining robust performance while reducing computational costs for large models.
>
> -   We extended DAPA to multimodal models, achieving a 24.27% DSR improvement on LLaVA-1.5-7B, showcasing its potential for aligning diverse architectures efficiently.
>
> **Reviewer EaeF**
>
> -   We addressed the concern about variability in knowledge measurement tools by planning to integrate tools like ROME for cross-validation, as shown in Appendix F, where our findings align with memory searching results.
>
> -   We acknowledged the trade-off between safety alignment and general capabilities, emphasizing that DAPA targets specific memory regions to minimize disruptions, with experiments showing only minor performance degradation while achieving significant safety improvements.
>
> -   We clarified that teacher models used for alignment are pre-trained using resource-intensive methods like RLHF and SFT, and DAPA complements these by reducing computational demands while exploring recognition-based refusal mechanisms for future enhancements.
>
> -   We discussed the similarities and distinctions between knowledge editing and unlearning, noting that unlearning focuses on erasing knowledge, while DAPA aligns specific knowledge without compromising the broader model competencies, as elaborated in the revised paper.
>
> -   We acknowledged the limitation of relying on pre-aligned teacher models, noting the lack of alternative alignment strategies in the current framework, and included this in the limitations section, with plans to explore self-supervised or rule-based approaches in future work.

---

> > ### Author Response · Authors · 2024-11-20
> > **General Rebuttal/Revision Response - Continued**
> >
> > **Reviewer Z3rx**
> >
> > -   We clarified that DAPA complements existing alignment methods like SFT/RLHF by enabling efficient transfer of alignment knowledge from a single aligned model to multiple unaligned models. While DAPA requires an aligned teacher model, it significantly reduces the computational cost of aligning additional models, addressing scalability challenges in safety enhancement.
> >
> > -   We acknowledged the limitation of relying solely on AdvBench for safety evaluation and expanded our experiments to include HarmfulQA and JailbreakBench, demonstrating consistent DSR improvements across diverse adversarial datasets.
> >
> > -   We clarified that the claim regarding FFN layers as primary storage for alignment knowledge is supported by additional experiments using multiple harmful prompts, showing consistent results across different scenarios.
> >
> > -   We conducted additional experiments using ShareGPT data, fine-tuning Llama2-7B with QLoRA, and found that DAPA improves DSR from 10.16% to 18.4% on AdvBench. Also, in our original experiments, we used the same chat template for both pre- and post-DAPA models, ensuring a fair comparison of performance and safety. This consistent setup guarantees that the improvements observed are directly attributed to our alignment method, not differences in model architecture or templates.
> >
> > -   We tested DAPA against advanced jailbreak tools like GPTFuzzer and demonstrated its robustness, with the DSR of Llama-2-7B increasing from 19% to 31%, confirming its adaptability to sophisticated adversarial attacks.
> >
> >
> >
> >
> >
> >
> >
> > Thank you again for your review!
> >
> > Best regards,
> >
> > Authors

---

### Author Response · Authors · 2024-11-27
**General Rebuttal/Revision Response - Continued**

We thank `Reviewer Z3rx` for providing valuable suggestions on additional experiments, which helped enhance the competitiveness of our paper. As a result, we have updated our paper with a new version. In case the reviewer requires a new categorisation for our paper revision, we have posted the updated revision here.

**Revised Appendix Mapping:**
- **New HarmBench Dataset experiment results in Appendix G.8**

	- Conducted an ablation study to evaluate 321 out-to-date harmful questions from HarmBench.

	- **Results**: Observed a 4.16% DSR improvement, confirming the method's robustness against challenging datasets.
- **New Advanced Jailbreak Adversary Testing GCG in Appendix G.11**.

    -   Conducted an ablation study to test to Llama-2 family models with GCG
    -   **Results**: DSR increased significantly, from 29.2% to 38.83%, showcasing effectiveness against sophisticated jailbreak adversarial. These results confirm that our method is dataset-agnostic and demonstrates robust performance on diverse and advanced evaluation scenarios.

- **New Advanced Jailbreak Adversary Testing AutoDAN in Appendix G.12**.

    -   Conducted an ablation study to test to Llama-2 family models with AutoDAN
    -   **Results**: DSR increased significantly, from 18.69% to 30.1%, showcasing effectiveness against sophisticated jailbreak adversarial. These results confirm that our method is dataset-agnostic and demonstrates robust performance on diverse and advanced evaluation scenarios.

- **Revise Advanced Jailbreak Adversary Testing AutoDAN in Appendix G.10**.

    -   Conducted an ablation study to test to Llama-2 family models with GPTFuzzer
    -   **Results**: DSR increased significantly, from 7.97% to 24.59%, showcasing effectiveness against sophisticated jailbreak adversarial. These results confirm that our method is dataset-agnostic and demonstrates robust performance on diverse and advanced evaluation scenarios.

-  **New Memory Space Analysis in Appendix G.15**

    -   Conducted an ablation study to provide additional visualization results of the memory space, incorporating 12 new visualizations in the updated version.

    -   **Results**: Hidden states in the middle layers significantly influence model output, with MLP layers exerting a higher indirect effect compared to attention layers. These findings are consistent with observations in Figure 3, reinforcing the critical role of middle-layer dynamics in model performance.

- **New Experiment on LLaVA-1.5-7B Multimodal Model in Appendix ~~G.8~~ G.9.**
- **New Advanced Jailbreak Adversary Testing in Appendix ~~G.9~~ G.10.**
- **New Experiment on LLaMA3-70B Model in Appendix ~~G.10~~ G.13.**
- **New Experiment on Fine-tuned Large Language Model in Appendix ~~G.11~~ G.14.**
- **New Memory Space Analysis in Appendix ~~G.12~~ G.15.**
- **New Appendix ~~G.14~~ G.16**: Example of DAPA on LLava-1.5 model.
- **New Appendix ~~G.13~~ G.17**: Model name and corresponding labels.

Thank you again for the review. We are writing to kindly ask if you have any additional comments regarding our response. In addition, if our new experiments address your concern, we would like to kindly ask if you could consider raising the score.

---

### Note · Authors · 2024-11-29

I have read and agree with the venue's withdrawal policy on behalf of myself and my co-authors.